# The somatically generated portion of T cell receptor CDR3α contributes to the MHC allele specificity of the T cell receptor

Philippa Marrack[1,2,3]*, Sai Harsha Krovi[3], Daniel Silberman[2,3], Janice White[2], Eleanor Kushnir[2], Maki Nakayama[3,4], James Crooks[5], Thomas Danhorn[5], Sonia Leach[2,5], Randy Anselment[5], James Scott-Browne[6], Laurent Gapin[3], John Kappler[1,2,3]

[1]Howard Hughes Medical Institute, Denver, United States; [2]Department of Biomedical Research, National Jewish Health, Denver, United States; [3]Department of Immunology and Microbiology, University of Colorado School of Medicine, Aurora, United States; [4]Barbara Davis Center for Childhood Diabetes, University of Colorado School of Medicine, Aurora, United States; [5]Division of Biostatistics and Bioinformatics, National Jewish Health, Denver, United States; [6]La Jolla Institute for Allergy and Immunology, La Jolla, United States

**Abstract** Mature T cells bearing αβ T cell receptors react with foreign antigens bound to alleles of major histocompatibility complex proteins (MHC) that they were exposed to during their development in the thymus, a phenomenon known as positive selection. The structural basis for positive selection has long been debated. Here, using mice expressing one of two different T cell receptor β chains and various MHC alleles, we show that positive selection-induced MHC bias of T cell receptors is affected both by the germline encoded elements of the T cell receptor α and β chain and, surprisingly, dramatically affected by the non germ line encoded portions of CDR3 of the T cell receptor α chain. Thus, in addition to determining specificity for antigen, the non germline encoded elements of T cell receptors may help the proteins cope with the extremely polymorphic nature of major histocompatibility complex products within the species.
DOI: https://doi.org/10.7554/eLife.30918.001

*For correspondence:
MarrackP@NJHealth.org

**Competing interests:** The authors declare that no competing interests exist.

## Introduction

Many T lymphocytes in the body express clonally distributed T cell antigen receptors composed of alpha and beta chains (TCRs) that react with peptides derived from pathogens and other foreign materials bound in a groove on the surface of host major histocompatibility proteins (MHCs) (*Allison et al., 1982*; *Babbitt et al., 1985*; *Haskins et al., 1983*; *Meuer et al., 1983*; *Shimonkevitz et al., 1983*). The genes encoding these MHC proteins are the most polymorphic genes in a given species. Most of the polymorphisms tend to be concentrated within the residues that line the peptide-binding groove of the molecules (*Bjorkman et al., 1987*). Hence, in general, different MHC alleles within a species preferentially bind, and present to TCRs, different peptides from any given invading organism. Thus the pathogen is unlikely to mutate such that none of its peptides bind to any of the MHC proteins expressed within the target species and the immune responses of at least some individuals within the infected species will be able to deal with the invading pathogen.

Many years ago another consequence of MHC polymorphisms was recognized. The allelic variants of MHC expressed in one individual are very frequently recognized by 1% or more of the T cells of other individuals expressing different MHC alleles, a phenomenon called 'alloreactivity'. While differences in bound peptides play an important role in alloreactivity (*Hunt et al., 1990*; *Crumpacker et al., 1992*), structural studies show that some of the allelic variations in MHC proteins themselves interact with the TCRs of alloreactive T cells (*Grandea and Bevan, 1993*; *Archbold et al., 2008*; *Colf et al., 2007*).

Experiments have shown that T cells in one individual are more likely to react with foreign peptides bound to the grooves of self MHC than to foreign peptides bound to foreign MHC (*Fink and Bevan, 1978*; *Zinkernagel et al., 1978*; *Kappler and Marrack, 1978*; *Sprent, 1978*). This phenomenon, known to be the consequence of thymic positive selection, is caused by the fact that thymocytes are allowed to develop into mature T cells only if the TCR they bear reacts with low affinity/avidity with MHC proteins bound to self peptides in the thymus (*Sprent et al., 1988*; *Ashton-Rickardt et al., 1994*; *Sebzda et al., 1994*; *Hogquist et al., 1994*). Paradoxically, in an apoptotic process termed 'negative selection', the thymus generally weeds out T cell progenitors that react with too high affinity/avidity with self MHC plus self peptide, thus preventing the maturation of many potentially self reactive T cells (*Kappler et al., 1987*; *von Boehmer et al., 1989*). Thus the collection of TCRs on mature T cells in any individual bears the footprint of positive selection, reacting almost undetectably with self MHC bound to self peptide and being more likely to react with foreign peptides bound to alleles of MHC to which they were exposed in the thymus than to peptides bound to unfamiliar MHC (*Fink and Bevan, 1978*; *Zinkernagel et al., 1978*; *Kappler and Marrack, 1978*; *Sprent, 1978*; *Hünig and Bevan, 1981*).

The simplest explanation for the effects of positive selection on the reactivity of mature T cells is that the phenomenon involves interactions between TCRs and allele-specific amino acids of MHC in the thymus. However, since different MHC alleles will bind different self peptides, positive selection may instead or, in addition, depend on interactions between TCRs and the MHC-bound self peptides. These ideas make different predictions about the portions of TCRs contributing to positive selection.

Mutational and structural studies have shown that the alpha and beta chains that comprise TCRs each usually engage MHC + peptide via three complementary determining loops (CDRs 1,2 and 3) (*Garcia et al., 1996*; *Reinherz et al., 1999*; *Dai et al., 2008*). For both the TCR alpha and beta chains, two of these loops, CDR1 and CDR2, are encoded by the germ line *Trav* (for the TCRα chain) and *Trbv* (for the TCRβ chain) genes. The third, CDR3, loop for each chain, on the other hand, is produced during TCR gene rearrangement as the cells develop in the thymus (*Davis, 1985*). Thus, the sequence coding for CDR3α, for example, is created when one of many *Trav* gene segments rearranges to fuse with one of the many *Traj* gene segments with the total number of possible CDR3α sequences increased by removal and/or addition of bases at the joining points of *Trav* and *Traj* (*Gellert, 2002*; *Cabaniols et al., 2001*; *Moshous et al., 2001*; *Lu et al., 2008*). This process creates the DNA coding for the entire Vα domain. The stretch of DNA coding for CDR3β is constructed along the same lines, by joining of one of a number of *Trbv*, *Trbd* and *Trbj* gene segments, again with bases removed or introduced at the joining points to form the CDR3 loop of the complete Vβ domain.

The fact that the TCR CDR1 and CDR2 loops are germline encoded and therefore relatively fixed, whereas the TCR CDR3 loops are at least partially somatically generated and therefore very variable led investigators to suggest that the CDR1 and CDR2 loops would contact germline encoded MHC whereas the CDR3 loops would contact the extremely variable and unpredictable foreign peptide. Indeed evidence that the CDR3 loops contact peptide rapidly appeared (*Danska et al., 1990*; *Kelly et al., 1993*; *Wither et al., 1991*). Other studies investigated the orientation of the TCR on MHC and suggested that the TCR might always lie approximately perpendicularly on MHC (*Jorgensen et al., 1992*) and that TCR/MHC interactions would always have the same orientation (*Sant'Angelo et al., 1996*). However, when crystallographically solved structures of TCRs on MHC became available it was soon apparent that TCRs are usually oriented diagonally on the MHC, but the angle of their interaction varies quite considerably from one structure to another. Moreover, the solved structures also showed that the predictions about contact points between CDR loops and MHC and peptide are by no means absolute. Although the TCR CDR3 regions are often focused on the peptide, amino acids in these regions sometimes also contact MHC and, vice versa, CDR1 and

CDR2 amino acids sometimes contact peptide in addition to their predicted interactions with MHC (*Garboczi et al., 1996*; *Garcia et al., 1996*; *Hennecke and Wiley, 2001*; *Meuer et al., 1983*; *Rudolph et al., 2006*).

These results bear on our understanding of positive selection in the thymus. Were positive selection to depend only on TCR/MHC interactions, and CDR1 and CDR2 to react only with MHC amino acids, one might predict that positive selection selects TCRs that react well with peptides bound to self rather than foreign MHC by picking out TCRs bearing TRAVs and TRBVs that react favorably with self MHC. Indeed there is evidence that this is the case (*Pircher et al., 1992*; *Merkenschlager et al., 1994*; *Sim et al., 1996*). Conversely, if positive selection were to depend only on TCR/self peptide interactions, as suggested by some studies (*Ignatowicz et al., 1996*; *Tourne et al., 1997*; *Nikolić-Zugić and Bevan, 1990*; *Hogquist et al., 1994*; *Ashton-Rickardt et al., 1994*; *Wong and Rudensky, 1996*; *Barton et al., 2002*), and CDR3 loops to react only with the MHC-bound peptide, then CDR3 regions might be the determining factor. However, as discussed above, amino acids in CDR1s and CDR2s sometimes react with the presented peptide and CDR3 amino acids can interact with MHC. Understanding of this issue is complicated by the cooperative nature of TCR interactions with its ligands, by which an interaction at one site on the TCR/MHC/peptide surface adjusts interactions elsewhere (*Mazza et al., 2007*; *Baker et al., 2012*; *Adams et al., 2016*) and a study that indicated that the entire sequence of the TCRα chain, including the TRAV, TRAJ and CDR3α, is involved in positive selection (*Merkenschlager et al., 1994*).

We set out to resolve these issues. We analyzed the TCRα repertoires of naïve CD4 T cells in mice that each expressed one of two TCRβ chains, DOβWT or DOβ48A (*Scott-Browne et al., 2009*), and a single MHCII protein, IA, of alleles b, f or s (for simplicity and ease of reading we will use IA to describe what are often termed I-A proteins, and we will not use superscripts to denote MHC and IA alleles). As predicted by previous studies (*Pircher et al., 1992*; *Merkenschlager et al., 1994*; *Sim et al., 1996*), the frequency with which mature T cells used different TRAVs was indeed affected to some extent by the MHCII allele on which they were positively selected and by the coexpressed TCRβ. Likewise the TRAJs used were affected by the selecting MHCII allele and coexpressed TCRβ, but demonstrated unexpected biases towards use of the TRAJs that were furthest from the TRAV locus.

Most surprisingly, however, the CDR3α sequences differed markedly depending on the MHCII allele and partner TCRβ in the mouse. This was true even if we compared, between MHC alleles, the TCRα sequences constructed from rearrangements involving the same TRAVs and TRAJs, indicating that the non germ line encoded portions of CDR3α are involved in MHCII allele specific selection.

## Results

### The generation and properties of mice expressing a single TCR beta chain

The impact of positive selection on the TCR repertoire of mature T cells cannot be understood by sequencing only the expressed *Tcra* or *Tcrb* chain genes. This is because others and we have found a fairly high percentage of individual TCRα or TCRβ sequences are expressed in animals regardless of their MHC haplotype (*Robins et al., 2010*; *Warren et al., 2011*; *Liu et al., 2014*) (*Supplementary file 1*). Presumably this is at least in part possible because each individual chain is paired with a different partner(s) in animals with different MHC alleles. Therefore, the pairs of TCR chains expressed in individual T cells must be known in order to understand the impact of thymus selection on the TCR repertoire of mature T cells.

The T cells in any given mouse or human have been reported to bear, collectively, more than $10^5$ different TCRα and about the same number of different TCRβ chain sequences (*Venturi et al., 2011*; *Li et al., 2016*). Thus the T cells might bear up to $10^{10}$ different combinations of these chains. Although methods for sequencing and accurately pairing the TCRα and TCRβ (or immunoglobulin heavy and light chain) RNAs from many individual T (or B) cells have been described (*Tan et al., 2014*; *DeKosky et al., 2013*), in our experience (*Munson et al., 2016*) these are still not able to cope with the large numbers of individual chains and combinations we expect in normal animals. Therefore we decided to limit our analyses to the T cells in animals that expressed a single TCRβ and any possible TCRα. This choice has two advantages. It allowed accurate knowledge of the TCRβ

on the T cells and, because it was expected that only a limited number of TCRα chains can be positively selected with a single TCRβ, it limited the numbers of different TCRαsequences we expected to find in the mice (*Merkenschlager et al., 1994*; *Fukui et al., 1998*; *Hsieh et al., 2006*).

We chose two TCRβ chains for these experiments. These were the TCRβ originally isolated from a T cell hybridoma constructed from BALB/c T cells specific for IA$^d$ or IA$^b$ bound to a peptide from chicken ovalbumin, the DOβWT TCRβ (*White et al., 1983*) and the same TCRβ with a mutation in its TRBV region such that the tyrosine at position 48 was changed to an alanine, DOβ48A (*Scott-Browne et al., 2009*). This mutation reduces the ability of the TCRβ chain to react with the alpha chain alpha helix of MHCII and with the alpha1 alpha helix of MHCI. The chain was chosen for our analyses because we thought that the TCRα sequences that could successfully overcome the deficits in MHC recognition by the TCRβ chain might more clearly illustrate the properties of the TCRα needed for successful positive selection.

The goal of these studies was to find out how the allele of MHC involved in thymic selection affects the sequences of the TCRs on the selected T cells. To achieve this we studied TCRs on naïve CD4 T cells that had been selected in some of the readily available mice that expressed a single MHCII protein, IAb, IAf and IAs (*Mathis et al., 1983*). Transgenic mice that expressed either DOβWT or DOβ48A and no other TCRβ were crossed such that they each expressed one of these MHCII alleles. The numbers of mature CD4 and CD8 T cells in the thymuses of the H2 b, f or s strains of mice were measured. As predicted by our previous data using retrogenic mice (*Scott-Browne et al., 2009*), the numbers of mature CD4 or CD8 thymocytes in mice expressing DOβ48A were much lower than those in mice expressing DOβWT (*Figure 1A*, *Figure 1—source data 1*). This was true regardless of the MHC allele on which the cells were selected. Thus the TCRβ 48A for 48Y substitution affects MHC interactions regardless of the MHC class or allele, as we have previously predicted (*Scott-Browne et al., 2009*). The difference in numbers of mature T cells between mice expressing DOβWT and DOβ48A was less marked in peripheral lymph nodes than in the thymus (*Figure 1B*), probably because of increased homeostatic expansion, as exemplified by the increased percentages of CD44hi T cells amongst those few that could mature in DOβ48A mice (*Figure 1C*).

There were more mature CD4 than CD8 T cells in the lymph nodes of mice expressing DOβWT and H2b (*Figure 1D*). The effect was much less marked in mice expressing H2f or H2s. The phenomenon may be due to the fact that DOβWT was found in a TCR that reacts with IAd or IAb plus a foreign peptide (OVA 327–339) and not from an MHCI-reactive TCR (*White et al., 1983*). The bias towards CD4 versus CD8 T cells in H2b animals was not manifest in lymph node cells bearing DOβ48A and was, indeed, reversed in animals expressing that TCRβ and H2f or H2s.

## The TCRα confers a bias towards reactivity with the selecting MHC allele

Mature T cells do not usually react detectably with self MHC alleles plus self peptides, the reactivity that presumably allowed their positive selection in the thymus. However, the potential inadequacies of DOβ48A allowed us to test whether or not the TCRαs that, on mature T cells, paired with it did indeed react preferentially with the MHCII allele on which they were positively selected. We guessed that introduction of the more prominently MHC-reactive DOβWT chain into DOβ48A T cells might reveal the underlying reactivity of the TCRα sequences in these T cells for various MHC alleles. Thus we isolated CD4 T cells from mice expressing the DOβ48A transgene, stimulated them with anti-TCR, transduced them with a GFP+ retrovirus expressing the DOβWT chain, and tested the ability of the transductants to react with cells expressing different alleles of MHC. CD69 expression was used as a marker of activation. Non transduced (GFP-) cells in the same cultures were used as controls. In no case did the nontransduced cells show a significant response. However, some of the DOβWT transduced cells responded. Notably, the percentage of the transduced cells that responded to challenge was always greatest if the antigen presenting cells expressed the MHC allele on which the T cells were positively selected *Figure 2*, (*Figure 2—source data 1*). For example, T cells from a DOβ48A H2b mouse, after transduction to express DOβWT, were most likely to react with H2b presenting cells and DOβWT transduced cells from DOβ48A H2s mice reacted only with challenge cells expressing H2s. These experiments show that the TCRα chain that pairs with the transgenic DOβ48A does indeed contribute to the preference of CD4 T cells to react with peptides bound to the MHCII allele involved in positive selection.

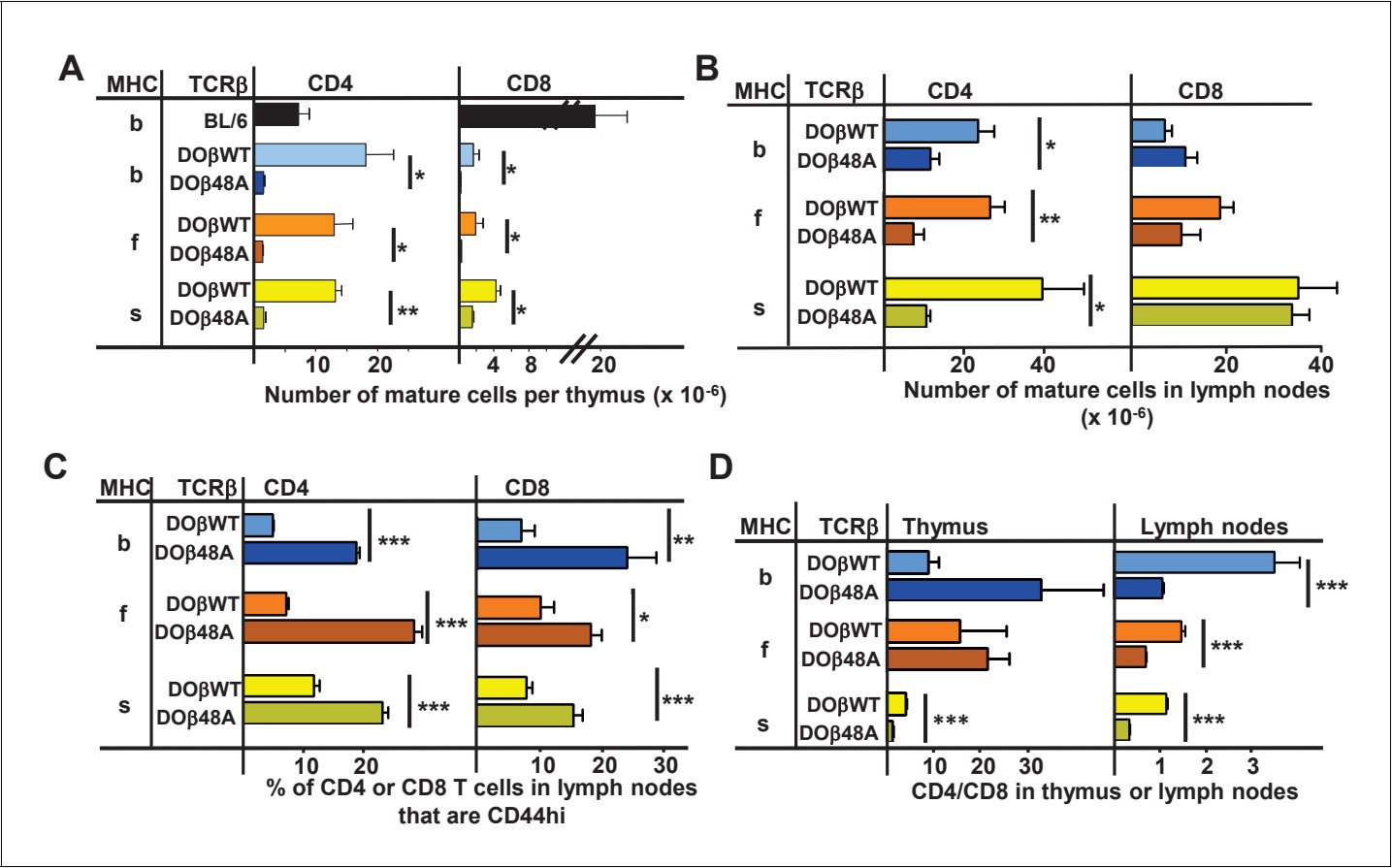

**Figure 1.** CD4 selection in mice expressing single TCRβ chains and different MHC alleles. Cells were isolated from the thymuses and lymph nodes of mice expressing a singl;e TCRβ, DOβWT or DOβ48A, and different MHC haplotypes and stained for expression of CD4 and CD8 and CD44. Results are the means and standard errors of the mean (SEMs) of three independently analyzed mice expressing the indicated TCRβs and MHC II alleles. Student t analyses were used to compare results between the DOβWT and DOβ48A paired samples. *p<0.05., **p<0.01, ***p<0.001.

DOI: https://doi.org/10.7554/eLife.30918.002

The following source data is available for figure 1:

**Source data 1.** Data from individual mice show that both CD4 and CD8 T cells appear in mice expressing a single TCRb chain regardless of the MHC allele expressed.

DOI: https://doi.org/10.7554/eLife.30918.003

## Expression of only one TCRβ chain limits the numbers of TCRα sequences that can participate in positive selection

Because allelic exclusion of the TCRα locus is not perfect (*Malissen et al., 1992*), mature T cells may express two functional TCRα proteins. To be sure that the TCRα chains analyzed in our experimental mice were actually those involved in positive selection of the cells bearing them, we crossed the DOβWT or DOβ48A transgenic, TCRβ-/- mice with TCRα-/- TCRβ-/- animals of each MHC haplotype to generate animals that were DOβWT or DOβ48A transgenic, TCRβ-/-, TCRα+/-. Naïve CD4 T cells were isolated from the lymph nodes of these animals and cDNA coding for their TCRαs were sequenced as previously described (*Silberman et al., 2016*). PCR and sequencing errors in the germ line encoded portions of these sequences were corrected as described in the Materials and methods section. To deal with possible sequencing errors in the non germ line encoded portions of CDR3α, sequences that occurred only once in any given sequencing run were eliminated from further analysis. In fact this decision affected the conclusions of all the experiments show below only slightly. Conclusions from analyses that included all sequences, or that eliminated sequences that occurred with the lowest 5% frequency in each sample were similar (data not shown).

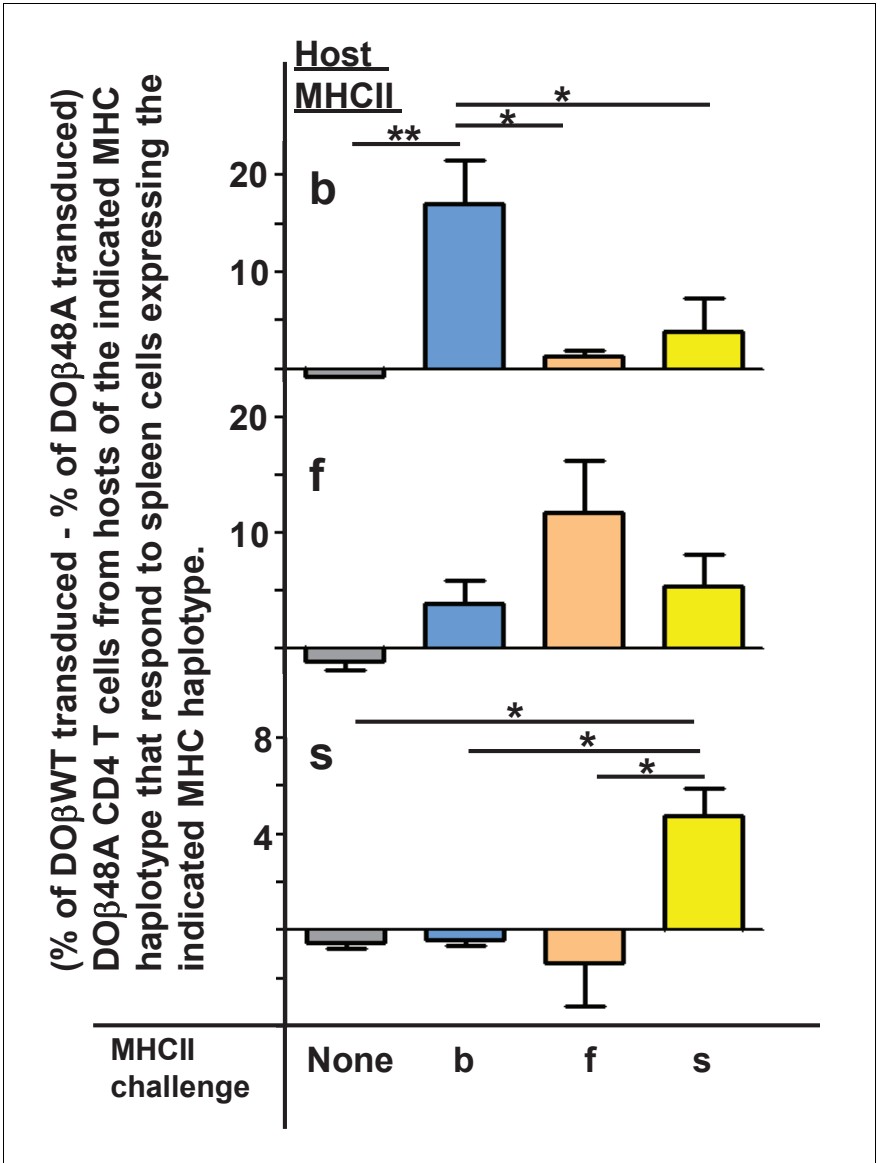

**Figure 2.** TCRα contributes to the MHCII allele bias of selected naïve CD4 T cells. Naïve CD4 T cells were isolated from the lymph nodes of DOβ48A H2b, f or s mice and incubated for 2 days in wells coated with anti-TCRβ and anti-CD28. Thus activated, the cells were spinfected with GFP-expressing retroviruses expressing also DOβWT or DOβ48A. The cells were cultured for a further 2 days and then challenged with spleen cells from mice expressing the indicated MHC alleles, or in the absence of added spleen cells. One day later the cells were stained for expression of CD69. Results were calculated as the (% of GFP+ T cells transduced with DOβWT-expressing retroviruses that were CD69+) – (the % of GFP+ T cells transfected with DOβ48A-expressing retroviruses that were CD69+) in wells containing the same challenge spleen cells. Shown are the average results ± standard error of the mean (SEM) from three independent experiments. *p<0.05, **p<0.01 by one way ANOVA followed by Neuman Keuls analyses.

DOI: https://doi.org/10.7554/eLife.30918.004

The following source data is available for figure 2:

**Source data 1.** After transduction with the DObWT chain, T cells from mice expressing DOb48A react with cells bearing the MHC allele that selected them.

DOI: https://doi.org/10.7554/eLife.30918.005

Others have previously reported that the T cells in mice expressing a single TCRβ chain have a limited repertoire of TCRα chains by comparison with WT animals (*Fukui et al., 1998*). To find out whether this applied to CD4 T cells expressing the DOβWT or DOβ48A TCRβ we constructed species accumulation curves for TCRα sequences in B6 and the TCRβ transgenic animals. These were performed by combining the TCRα sequences from all mice of the same genotype or by plotting the TCRα sequences for individual mice of each genotype (*Figure 3—figure supplement 1*, the source data for these and all subsequent figures are at GEO accession GSE105129). Species accumulation curves show that the total number of TCRα sequences we could detect on naïve CD4 T cells in the TCRβ transgenic animals ranged from less than 5000 to a maximum of about 30,000. These numbers are less than those found in CD4 naïve T cells from B6 mice, which we found to be similar in number to those found on mouse CD8 T cells (*Genolet et al., 2012*), >than $10^5$ in number (*Figure 3*). The numbers of TCRα sequences that could partner with DOβWT in selection of CD4 T cells varied considerably with the selecting MHCII allele. More than four times more TCRα sequences were apparent in mice expressing IAb versus IAs (*Figure 3B–D*), perhaps because the TCR from which DOβWT is derived can be selected by IAb (*Liu et al., 1996*). This effect of MHC allele on the numbers of selected TCRα chains was not evident in animals expressing DOβ48A. Notably, the numbers of different TCRα chains associated with DOβ48A was lower than those associated with DOβWT regardless of the MHCII allele involved, possibly because of the extra demands imposed on TCRα chains by the inadequate TCRβ chain lacking an important MHC contact residue, Y48 (*Scott-Browne et al., 2009*).

Perhaps the real surprise in these results is how many TCRα sequences can partner with a single TCRβ and participate successfully in positive selection since work in humans and mice suggest that, on peripheral T cells, each TCRβ partners with only about 5–25 different TCRαs (*Arstila et al., 1999*; *Casrouge et al., 2000*; *Venturi et al., 2011*; *Li et al., 2016*).

## Expressed TCRα sequences are strongly influenced by the selecting MHC allele and partner TCRβ

We compared the frequency with which particular TRAV/TRAJ/CDR3α amino acid sequences, that is, the entire TCRα sequences, occurred in the various strains of mice. Data of this type can be compared in several ways. The data can be analyzed to find out whether a particular TRAV/TRAJ/CDR3 sequence occurs in each sample, regardless of how often it appears in the set (comparison of unique sequence use). In this case, Jaccard similarity coefficients can be used to measure the similarity between samples. On the other hand, use of particular TRAV/TRAJ/CDR3 sequences can be compared taking into account the number of times a particular combination occurs. In this case Anne Chao Jaccard abundance based indices (*Chao et al., 2012*) are an appropriate statistical tool. Both methods were used in the comparisons shown in *Figure 4*. Jaccard analyses showed that the same combination of TRAV/TRAJ/CDR3 sequences were likely to appear in samples from mice of the same TCRβ and MHC genotype but were very unlikely to be shared with the T cells from mice expressing a different MHC allele (*Figure 4A*). This was just as apparent when the abundance with which the sequences were expressed was taken into account (*Figure 4B*). Thus these data show that, given a single TCRβ, the TCRα sequences that can participate in positive selection are dramatically affected by the selecting MHCII allele. Moreover, the fact that the values of the Anne Chao Jaccard analyses for mice of the same MHCII allele are much larger than those of the Jaccard analyses shows that sequences that appear frequently in one mouse of a given genotype are more likely to be found in other mice of the same type. Such a result is a manifestation of the fact that some sequences were repeated many times in all mice of a given MHCII type, whereas other sequences were rare. This uneven and certainly non-Poissonian distribution of TCR sequences has been observed before (*Correia-Neves et al., 2001*; *Fazilleau et al., 2005*; *Freeman et al., 2009*). The phenomenon was not necessarily caused by expansion of single clones of T cells since, even in sets in which the CDR3α amino acid sequences were identical, the DNA sequences were not necessarily all the same (data not shown).

Similar analyses were applied to samples from mice in which the selecting MHCII allele was identical, but the partner TCRβ differed. The data show that the selected TCRα chains depended on the partner TCRβ, even when the selecting MHC allele was identical. Close inspection revealed, however, that there was slightly more overlap if the co-selected TCRβs were different but the selecting MHCII alleles were identical than if the reverse were true, that is the co-selected TCRβs were the

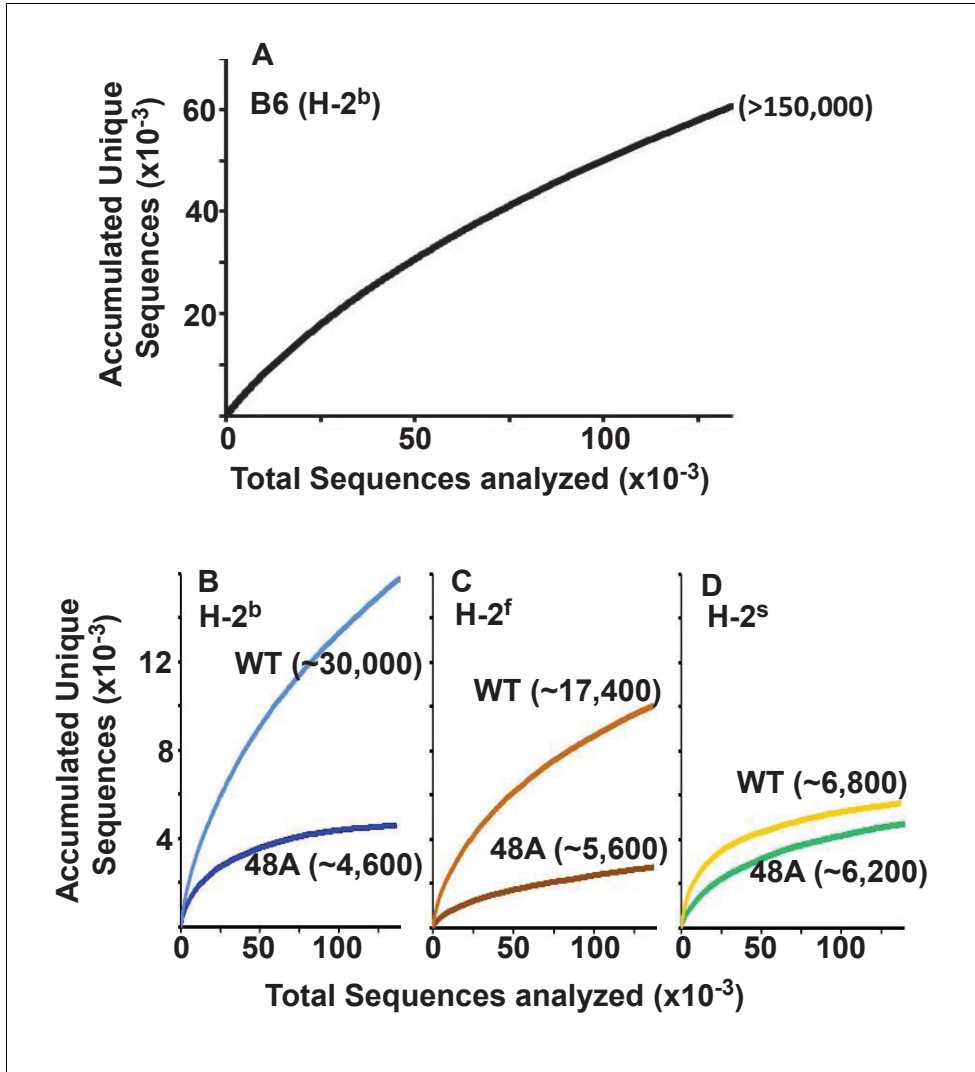

**Figure 3.** Expression of a single TCRβ chain, DOβWT and, even more markedly, DOβ48A, reduces the number of different TCRα chains that can be positively selected, regardless of the selecting MHCII allele. Naïve CD4 T cells were isolated from the spleens (B6) or lymph nodes of mice expressing MHC b, f or s, single TCRβ chains and heterozygous for expression of functional TCRα chains. Their expressed TCRα chains were sequenced and analyzed with species accumulation curves. Results were combined from three independently sequenced data sets from mice of each genotype except for those for H2s DOβWT animals, which were combined from only two independently sequenced animals. Data are shown together with an estimate (bracketed) of the total numbers of different TCRα protein sequences present in the naïve CD4 T cells of each type of mouse.
DOI: https://doi.org/10.7554/eLife.30918.006

The following figure supplement is available for figure 3:

**Figure supplement 1.** The naïve CD4 T cells in mice expressing a single TCRa chain express a limited number of TCRα sequences regardless of the MHC allele involved in their selection in the thymus.
DOI: https://doi.org/10.7554/eLife.30918.007

same but the selecting MHCII alleles were different (*Figure 4C,D*, *Figure 4—figure supplement 1*) (*Fink and Bevan, 1978*).

A few sequences appear in at least one mouse of each haplotype. For example, 16 sequences appear in DOβWT mice expressing MHCII b, f or s (but not in any DOβ48A animals) (data not shown). Such sequences might belong to yet undiscovered types of T cells that express an invariant TCRα, like iNKT cells or MAIT cells (*Chandra and Kronenberg, 2015*; *Gapin, 2009*). We think this is unlikely to be true because, in the complete naïve CD4 T cell sequences, we did not consistently find

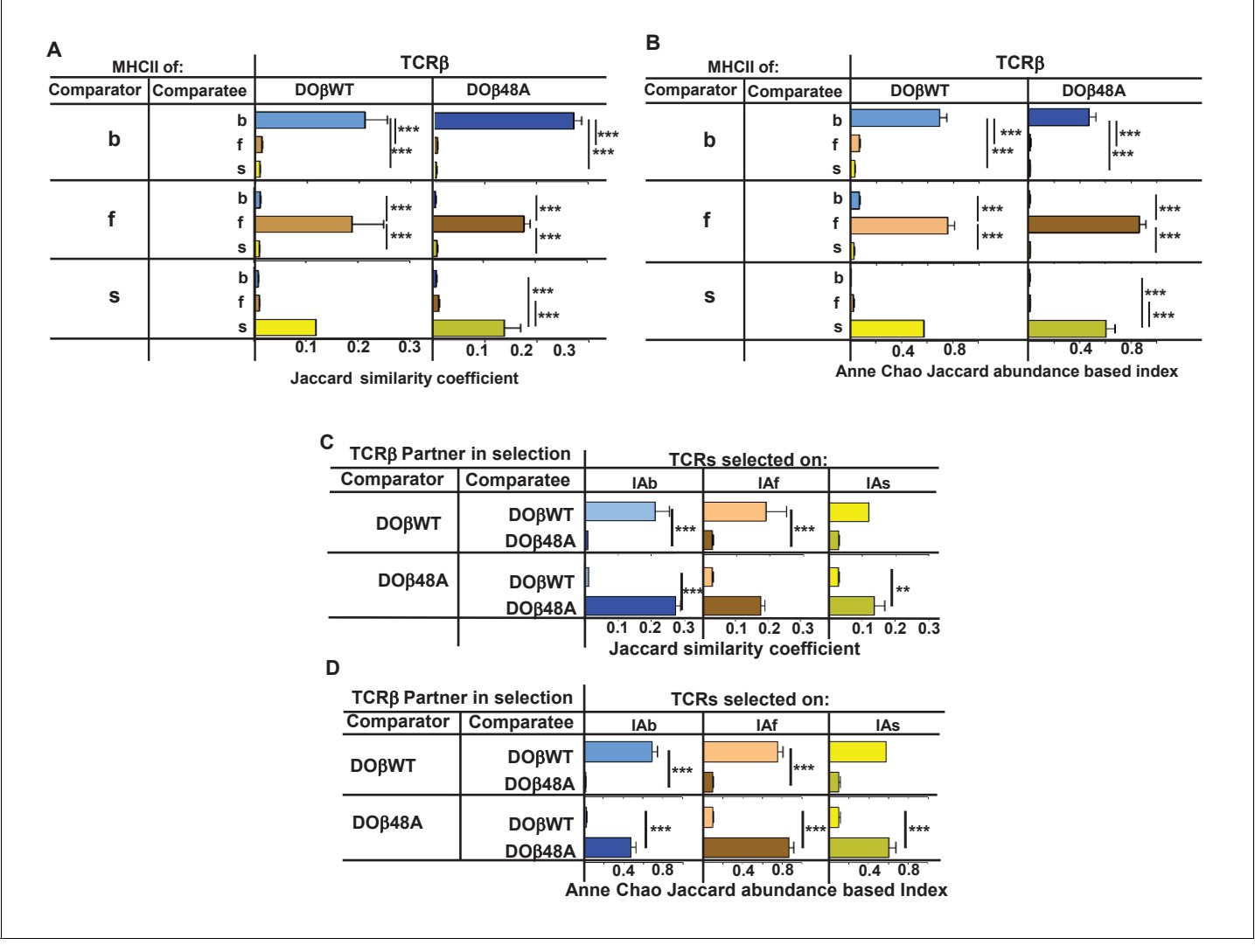

**Figure 4.** TCRβ sequences on naïve CD4 T cells are determined by the selecting MHCII allele and the co-selected TCRβ. TCRαs on naïve CD4 T cells from the lymph nodes of TCRα+/- mice expressing a single TCRα and various MHC alleles were sequenced and analyzed as described in *Figure 3*. Results are the means and SEMs of three independently sequenced animals of each genotype except for H2s DOβWT animals, of which only two mice were analyzed. ***p<0.001 by one way ANOVA with Newman-Keuls post analysis.

DOI: https://doi.org/10.7554/eLife.30918.008

The following figure supplement is available for figure 4:

**Figure supplement 1.** TCRα sequences are somewhat more likely to be shared between T cells selected on the same MHCII allele but differing in TCRβ than between T cells sharing TCRβ but selected on different MHCII alleles.

DOI: https://doi.org/10.7554/eLife.30918.009

the sequences of the iNKT cell or MAIT cell TCRαs. Probably this was because the cells bearing the iNKT cell or MAIT cell invariant TCRαs were in the activated/memory T cell populations, which were not examined in our experiments. Were there to be an undiscovered T cell subset bearing another invariant TCRα it would presumably also be in the activated/memory T cell population and therefore not included in our assays.

These data show that positive selection acts on CD4 T cell precursors, via the action of the expressed MHCII allele on particular TCRα/TCRβ pairs.

## TRAV usage depends on the selecting MHCII haplotype and the partner TCRβ chain

In order to find out which element(s) of TCRα determine MHC allele specificity we analyzed each element separately using data from the experiments described above. Others have previously reported that certain TRAVs are used more frequently by CD4 versus CD8 T cells or in mice expressing particular alleles of MHC (*Jameson et al., 1990*; *Pircher et al., 1992*; *Sim et al., 1996*; *Simone et al., 1997*; *Merkenschlager et al., 1994*). *Trav* rearrangements occur in thymocytes after the cells have rearranged their TCRβ genes (*Lindsten et al., 1987*). Thus, the frequency with which TRAVs appear on mature naïve CD4 T cells is predicted to depend on a number of issues, the ease with which the TRAV gene can rearrange (*Chen et al., 2015*), its ability to pair with the preexisting TCRβ expressed in the cell (*Vacchio et al., 1993*) and the ability of the TCRα/TCRβ pair to participate in positive, but not negative, selection on the MHCII protein expressed in the thymus.

We first tested whether the expressed MHC haplotype might unexpectedly affect the nature of TRAVs expressed on preselection thymocytes. As shown in *Figure 5A*, *Figure 5—figure supplements 1–2*, TRAV usage on preselection thymocytes was similar, regardless of the MHC allele in the donor animal or the coexpressed TCRβ(s). There were, however, some interesting aspects of TRAV use on preselection thymocytes. . TRAVs whose genes are most proximal to the TRAJ locus (TRAVs 17–21) were frequently rearranged, as predicted by previous studies (*Villey et al., 1996*; *Shih et al., 2011*; *Genolet et al., 2012*) (note that the TRAVs are arranged by family and not by position in the TRAV locus). However, in preselection thymocytes we also observed frequent rearrangements involving TRAV 1 and members of the TRAV 3, 6, 7, 10, 11 and 14 families. In the cases of the TRAV families the frequently rearranged TRAVs were not always those which are most proximal to the TRAJ locus. For example, amongst the TRAV7 family, the most frequently rearranged member was TRAV7-2D, one of the family members that is furthest from the TRAJ locus, whereas the close relative of TRAV7-2D, TRAV7-2A, was not frequently rearranged. This suggests that chromatin structure, promoter accessibility and use by rearranging processes also play a role in TRAV rearrangements (*Chen et al., 2015*).

TRAV use by T cells from mice of the same genotype was very similar (*Figure 5B–G*, *Figure 5—figure supplements 2–3*). However, the selecting MHC allele affected the frequency with which different TRAVs were expressed on mature naïve CD4 T cells (*Figure 5—figure supplements 2–3*). For example members of the TRAV5 family were used to some extent by CD4 T cells selected on IAb, but not by cells selected on IAf or IAs (compare *Figure 5B,C* with *Figure 5D–G*). On the other hand, CD4 T cells in DOβWT H2s mice were alone in their use of TRAV17 (*Figure 5B,D and F*). Differential use of TRAVs was much more marked in DOβ48A mice and illustrated the TRAV preferences of mice selected on different MHCII alleles more strikingly. For example, DOβWT cells selected on IAs used most members of the TRAV6 family whereas DOβ48A expressing cells selected on the same MHC allele used, of the TRAV 6 family, almost entirely TRAV6-5D and TRAV6-7DN and also used more frequently than T cells selected on other MHCII alleles, members of the TRAV16 family. Perhaps this reflects a greater need for basic amino acids in TRAV CDR1 and CDR2 for selection of H2s with DOβ48A as a partner, since, of the TRAV6 family, TRAVs6-5D and TRAV6-7DN (and TRAV6-5A and TRAV6-7DN) have a total of two basic amino acids in these elements whereas other members of the family have none. Likewise all expressed members of the TRAV16 family contain 2 or 3 basic amino acids in their CDR1 and CDR2 segments. This narrowing in TRAV choice by DOβ48A cells may reflect the increasing demands for selection imposed by the absence of the tyrosine at position 48 of the TCRβ chain.

Our analyses are based on a method in which cDNAs from individual mouse T cells are amplified simultaneously with a reverse TRAC oligo and oligos built to match each TRAV family (see Materials and methods Section). Therefore the differences in TRAV discovery could be due to more efficient PCR amplification of some TRAV genes than others. However, since the efficiencies of detection will be similar between members of the same family, it is legitimate to compare the frequency of rearrangement between different members of the same family, or the frequency of use of the same TRAV in mature T cells selected on different MHCs or with different TCRβ partners (see below).

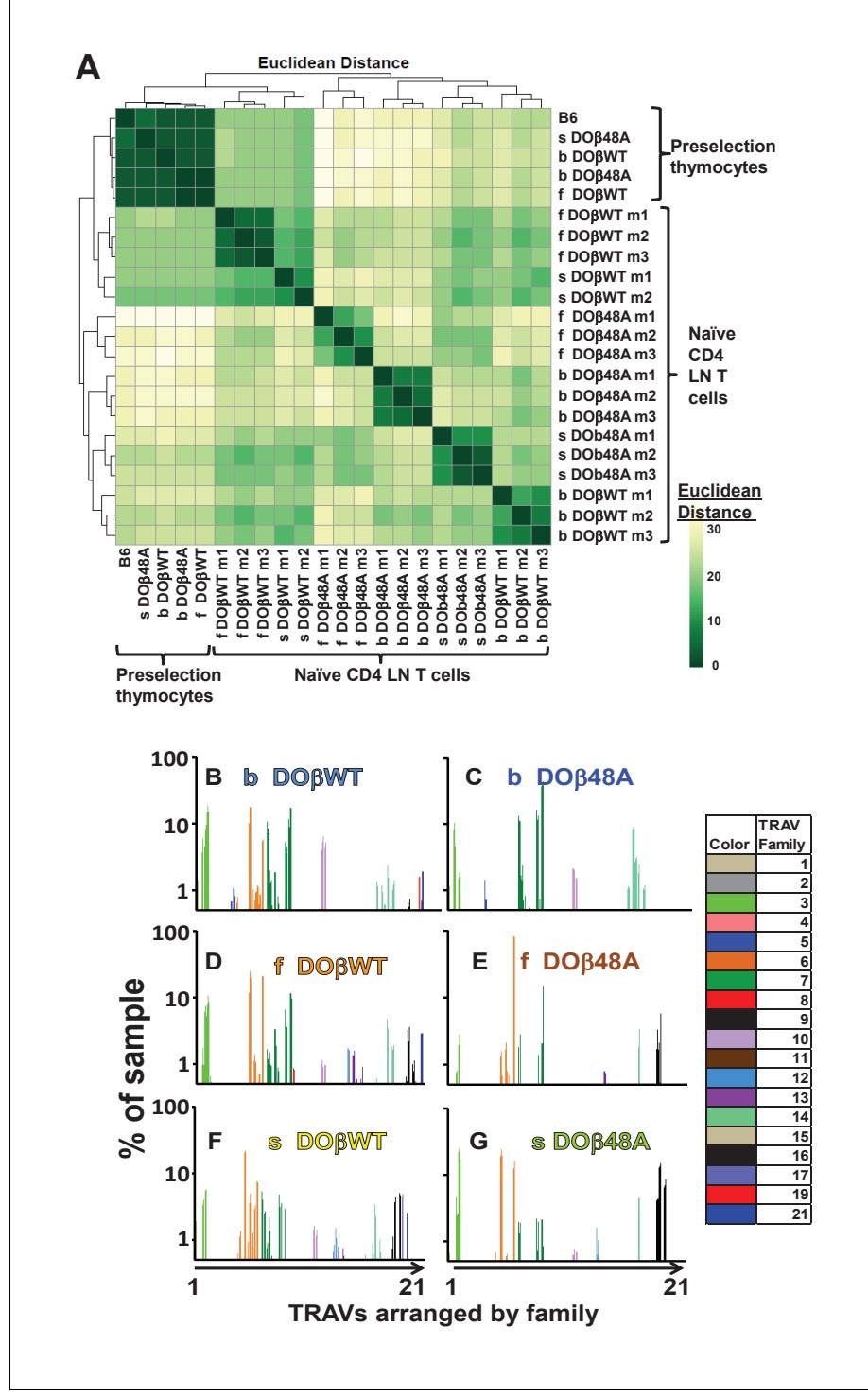

**Figure 5.** The frequency with which TRAVs are used on naïve CD4 T cells in TCRβ transgenic mice depends on their selecting MHCII and their partner TCRβ. TCRαs on preselection thymocytes or naïve T cells from the lymph nodes of TCRα+/- mice expressing a single TCRβ and various MHC alleles were sequenced and analyzed as described in *Figure 3*. (A) Shown are the Euclidean distances for TRAV use between the data for individual mice. Samples are hierachically ordered. Individual mice of the same genotype are numbered m1-3. (B) The average % use of each TRAV in mice expressing the indicated MHCII allele and TCRβ. Results are the means ± SEMs of 3 identical mice, except for H2s DOβWT animals, for which results are the averages of 2 mice. TRAVs are ordered by family, not by position on the chromosome.

*Figure 5 continued on next page*

*Figure 5 continued*

DOI: https://doi.org/10.7554/eLife.30918.010

The following figure supplements are available for figure 5:

**Figure supplement 1.** Different TRAVs are detected at different frequencies in preselection thymocytes.
DOI: https://doi.org/10.7554/eLife.30918.011

**Figure supplement 2.** TRAV usage by naïve CD4 T cells depends on the selecting MHCII allele and partner TCRβ.
DOI: https://doi.org/10.7554/eLife.30918.012

**Figure supplement 3.** TRAVs are used to different extents by naïve CD4 T cells in mice expressing different MHCII alleles and/or different TCRβs.
DOI: https://doi.org/10.7554/eLife.30918.013

## TRAJ usage depends on the selecting MHCII haplotype and the partner TCRβ chain

TRAJ use by preselection thymocytes was similar regardless of the selecting MHC haplotype or co-expressed TCRβ (*Figure 6A*, *Figure 6—figure supplements 1–2*). TRAJ use by naïve CD4 T cells from B6 mice was fairly uniform across the locus (*Figure 6B*). Unexpectedly, however, and in contrast to preselection thymocytes and naïve CD4 T cells from B6 mice, TRAJ use by T cells from mice expressing a single TCRβ was much more uneven and tended towards TRAJs whose genes were distal to the TRAV locus (*Figure 6B–L*). Regardless of MHC allele, CD4 T cells in DOβWT animals used TRAJ21 most frequently (*Figure 6D–F*). The reasons for this bias are unknown. TRAJ21 contains a tyrosine at or near the contact point with MHC but other TRAJs have a tyrosine similarly situated and they are not overexpressed. Moreover TRAJ21 is not overexpressed in T cells expressing DOβ48A, T cells that might be expected to be even more readily selected with an added tyrosine (*Scott-Browne et al., 2009*). The bias towards use of distal TRAJ genes was even more marked in animals expressing DOβ48A. In these mice TRAJ 9, TRAJ 12 and TRAJs 9,15 and 31 dominated in H2b, H2f and H2s mice respectively (*Figure 6G–I*). Pairwise comparisons between different mice are shown in *Figure 6—figure supplement 1* and DESeq 2 analyses are in *Figure 6—figure supplement 2*.

We do not know why the distal TRAJ genes were preferred in mice in which the TCRα repertoire was limited by the presence of a single TCRβ. In another study with a fixed TCRβ chain, a bias towards proximal TRAJs was noted with TRAV17, a TRAV that is close to the TRAJ locus (*Casanova et al., 1991*). The same publication described biases, depending on MHC allele towards use of Type 1 (G rich) or type 2 TRAJs. These explanations don't apply here, since in the experiments presented here TRAV expression was not particularly biased towards the distal TRAVs and the used TRAJs don't fall particularly into Types 1 or 2. It is possible that the choice is related to the DOβ chain itself. Alternatively it maybe that, because it is difficult for thymocytes expressing a single TCRβ to find a TCRα that can pair with the TCRβ and contribute to positive selection, multiple TCRα rearrangements have to occur in each thymocyte before a suitable TCRα partner is found. This will inevitably drive expressed TCRαs towards use of the distal TRAJs, although why these should satisfy the demands of positive selection more frequently than the proximal TRAJs do, is not obvious, at least from their amino acid sequences. It has recently been reported that prolonged expression of RAG protects cells, to some extent, from death (*Karo et al., 2014*). If the thymocytes in TCRβ transgenic mice have to express RAG for a longer time to find a suitable TCRα partner, then the prolonged expression of RAG needed for the multiple rearrangements required to access the TRAC proximal TRAJs might preferentially allow survival of the thymocytes in which this prolonged expression has occurred. Preliminary analyses of the naïve CD4 T cells in the various mice did not, however, suggest that the T cells in the TCRβ transgenic mice were more resistant to death than the equivalent cells in B6 animals. Finally, it is possible that, during the multiple rearrangements that may occur as TRAJ use moves to those at the distal portion of the TRAJ locus, in thymocytes the strength of signal received from the TCR/MHC/peptide interaction needed to drive positive selection might be reduced. Thus selection may occur more easily for thymocytes using distal rather than proximal TRAJs (*Moran et al., 2011*; *Seiler et al., 2012*). Future experiments will test this idea.

Overall, like the use of TRAVs, use of TRAJs depended on both the MHC haplotype and TCRβ present in the animals.

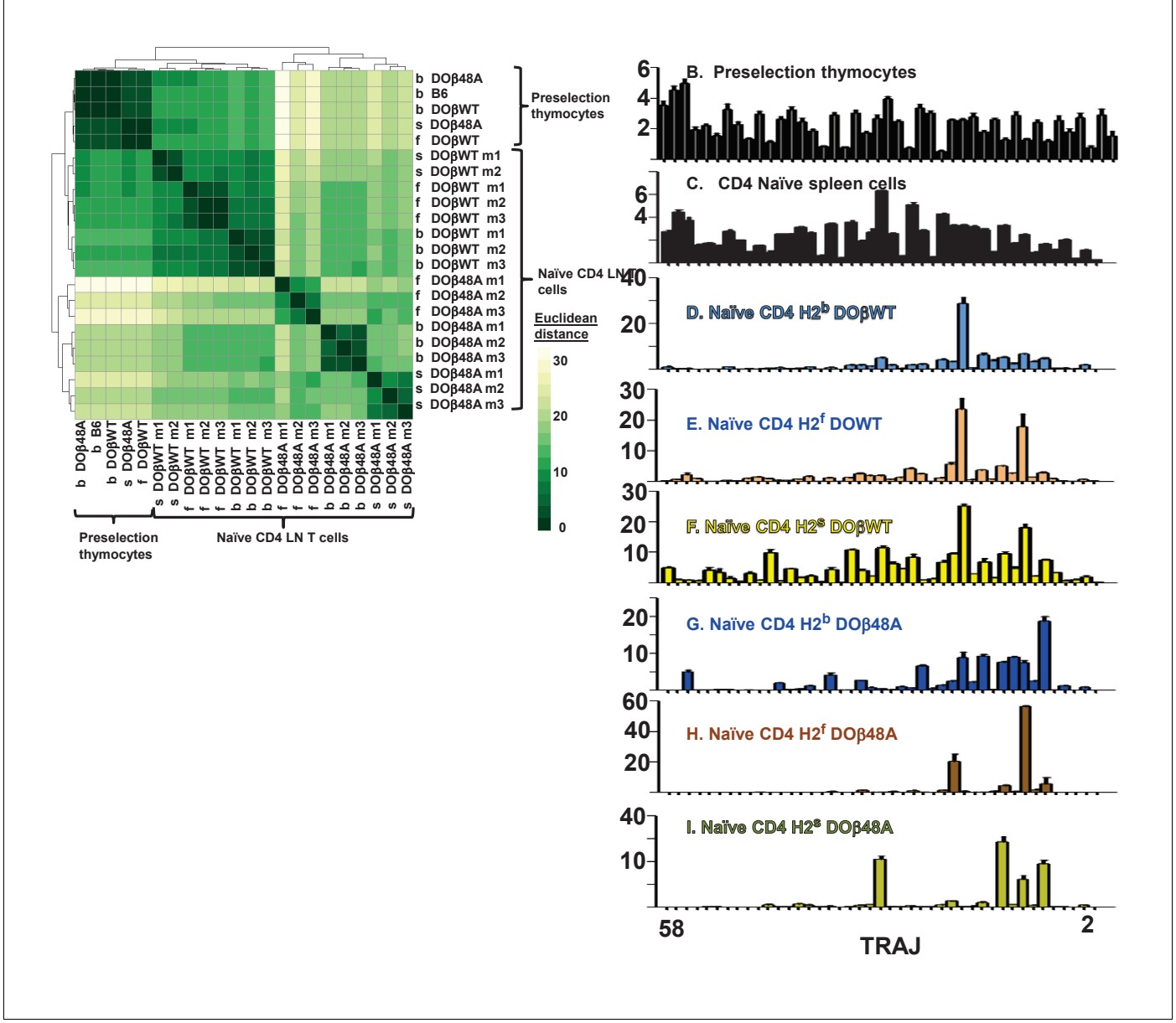

**Figure 6.** Naïve CD4 T cells in DOβWT or DOβ48A mice preferentially use TRAJs from the distal end of the TRAJ locus. TCRαs on preselection thymocytes or naïve CD4 T cells from the lymph nodes of TCRα±mice expressing a single TCRβ and various MHC alleles were sequenced and analyzed as described in *Figure 3*. (A) Shown are the Euclidean distances for TRAJ use between the data for individual mice. Samples are hierachically ordered. Individual mice of the same genotype are numbered m1-3. (B-L) The % use of each TRAJ in mice expressing the indicated MHCII allele and TCRβ. Results are the means and SEMs of 3 identical mice, except for H2s DOβWT animals, for which results are the averages of 2 identical mice. TRAJs are ordered by position on the chromosome. Also shown are the means and SEMs of TRAJ use by five independently sequenced preselection thymocytes and three independently sequenced naïve CD4 T spleen T cells from B6 mice.

DOI: https://doi.org/10.7554/eLife.30918.014

The following figure supplements are available for figure 6:

**Figure supplement 1.** TRAJ usage by naive CD4 T cells depends on the selecting MHCII allele and partner TCRβ.

DOI: https://doi.org/10.7554/eLife.30918.015

**Figure supplement 2.** TRAJs are used to different extents by naïve CD4 T cells in mice expressing different MHCII alleles and/or different TCRβs.

DOI: https://doi.org/10.7554/eLife.30918.016

## Expressed CDR3α sequences are strongly influenced by the selecting MHC allele and partner TCRβ

Because of the removal or introduction of bases when TRAVs rearrange to TRAJs, CDR3α protein sequences can vary in the number of amino acids they encode between the conserved C terminal cysteine of the TRAV and the conserved phenyl alanine/leucine/tryptophan glycine pair in the TRAJ region. We compared the average lengths of CDR3αs and the predicted number of N region bases between mice expressing the same TCRβ and different MHCII alleles. CD4 T cells selected on IAb had significantly shorter CDR3α lengths and fewer N region bases than their counterparts selected on IAf or IAs (*Figure 7—figure supplement 1*).

The analyses shown in *Figure 4* compared the frequencies with which entire TCRα sequences appeared under different selecting circumstances. We also analyzed how often particular CDR3α sequences are found in mice that differed in the selecting MHC allele or in the co-expressed TCRβ, using, again, Jaccard or Anne Chao Jaccard analyses to compare particular sequences without or with taking into account the abundance with which they occurred. Data comparing the occurrence of CDR3α protein sequences between mice that expressed the same or different MHCII alleles are shown in *Figure 7A–B*, and data comparing the occurrence of CDR3α protein sequences between mice expressing the same MHCII but different partner TCRβs are shown in *Figure 7C–D*. The results were similar to those obtained when comparing the entire TCRα sequences. The expressed CDR3α sequences in mice with a particular MHCII allele were very unlikely to be found in CD4 T cells of mice expressing a different MHCII allele, even if the co-selected TCRβ were the same in the mice (*Figure 7A–B*). Likewise, CDR3α sequences co-selected with a particular TCRβ were unlikely to be shared with those co-selected with a different TCRβ, even if the selecting MHCII allele were the same.

The first few amino acids of CDR3α (defined as the stretch between the last C of the TRAVs and the conserved F/W/L G sequence of the TRAJs) are encoded by the TRAVs themselves. Likewise, the last few amino acids of CDR3α are encoded by the TRAJs. Therefore the fact that the CDR3α sequences are controlled by the MHCII allele on which they were selected might have been, to some extent, dictated not by the non germline encoded amino acids in CDR3α but rather by the TRAV encoded amino acids downstream of the cysteine at the C terminal end of the TRAVs or by the TRAJ encoded amino acids upstream of their conserved TRAJ F/W/L. This problem applies particularly to the use of TRAVs since TRAV CDR1 and CDR2 amino acids may contact MHCII and thereby contribute to thymic selection whilst also dictating the first few amino acids of the accompanying CDR3 region.

We therefore checked whether CDR3α sequences associated with particular TRAV/TRAJ pairs differed between T cells selected on different MHCII alleles or associated with different TCRβs. Only a few of the possible TRAV/TRAJ pairs were present in sufficient numbers in all of the mice to be compared, so only a few such comparisons could be made. Examples of such comparisons are shown in *Figure 8*. Summaries combining all allowable results (in which all the TRAV.TRAJ combinations to be compared included least five different CDR3 sequences/mouse) are shown in *Figure 8—figure supplements 1–2*. T cells expressing DOβWT and the same TRAV and TRAJ combinations, but selected on different MHCII alleles or with different TCRβs clearly had CDR3α sequences that were almost completely unique to the selecting MHCII alleles.

A recent study has reported that thymocytes with aromatic/hydrophobic amino acids at the tips of their CDR3β segments are biased towards MHC reactivity, regardless of the selecting MHCII allele (*Stadinski et al., 2016*). The observations in the paper applied to CD4+ CD8+ (double positive thymocytes) that had been positively, but not negatively, selected, identified by their expression of CD69, and to regulatory T cells compared to preselection thymocytes. Such cells have not been examined in the experiments described here, so we cannot tell directly whether a similar observation applies to TCRα sequences. On the whole the evidence is that cells with aromatic amino acids at positions 6 and 7 on CDR3α are not particularly eliminated by clonal deletion in the thymus (data not shown). Nevertheless, we evaluated individual amino acids that would probably be at the tips of CDR3αs in CDR3α of different lengths. The results show MHCII allele and TCRβ specific selection for particular amino acids and also changes in amino acid preference at CDR3α positions depending on the length of the CDR3 (*Figure 8—figure supplement 2*). For example, arginine was very frequently used at position 4 in 12 amino acid long CDR3αs selected by IAs with DOβ48A, and similarly over

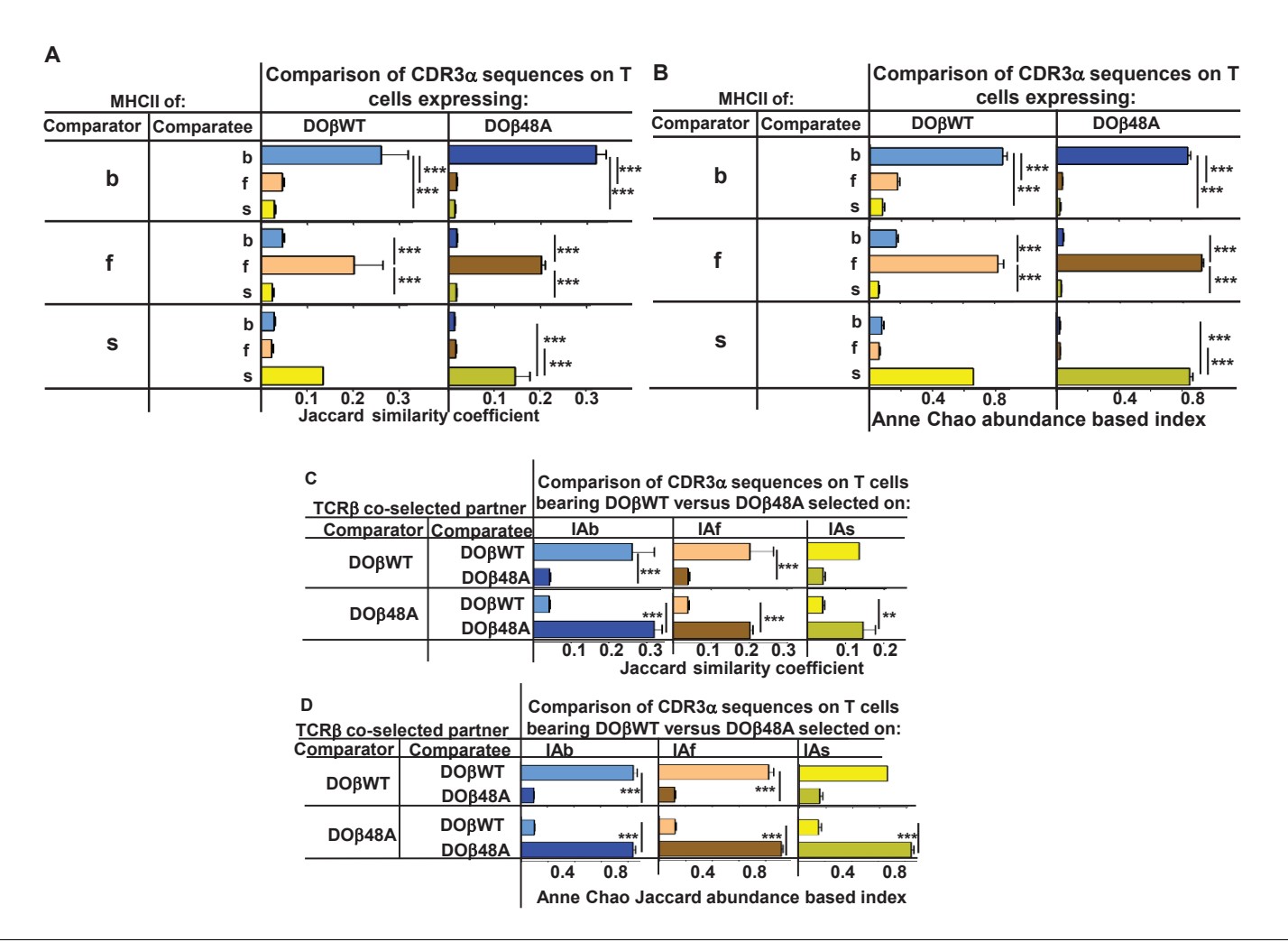

**Figure 7.** CDR3α sequences on naïve CD4 T cells are determined by the selecting MHCII allele and the co-selected TCRβ. TCRαs on naïve T cells from mice expressing a single TCRβ and various MHCII alleles were sequenced and analyzed for their CDR3α sequences as described in *Figures 3* and *4*. CDR3α sequences were defined as the amino acids between and including the conserved cysteine at the C terminal end of the TRAV and the conserved phenyl alanine, tryptophan or leucine in the TRAJ region. Shown are the means and SEMs of 3 independently sequenced identical mice except for H2s DOβWT mice, in which case only two mice were analyzed. ***p<0.001, **p<0.01 by one way ANOVA with Newman-Keuls post analysis.
DOI: https://doi.org/10.7554/eLife.30918.017
The following figure supplement is available for figure 7:

**Figure supplement 1.** CDR3α length on naïve CD4 T cells depends upon the selecting MHC allele and the co-selected TCRβ.
DOI: https://doi.org/10.7554/eLife.30918.018

selected at position 5 in 14 amino acid long CDR3αs selected on IAf with DOβWT, but much less frequently used by other MHC selection, TCRβ, CDR3α length combinations. Phenyl alanine was only used with evident frequency at position 5 in 14 amino acid-long CDR3αs selected on IAb with DOβWT. Apart from the phenyl alanine result there was no particular enrichment for aromatic amino acids at these tips.

## Discussion

It has long been known that T cells bearing αβTCRs are biased towards recognition of antigenic peptides bound to the allele of MHC to which the T cells were exposed in the thymus(*Fink and Bevan, 1978*; *Zinkernagel et al., 1978*; *Kappler and Marrack, 1978*; *Sprent, 1978*). This phenomenon, known as positive selection, has been ascribed to a requirement for a low affinity/avidity reaction

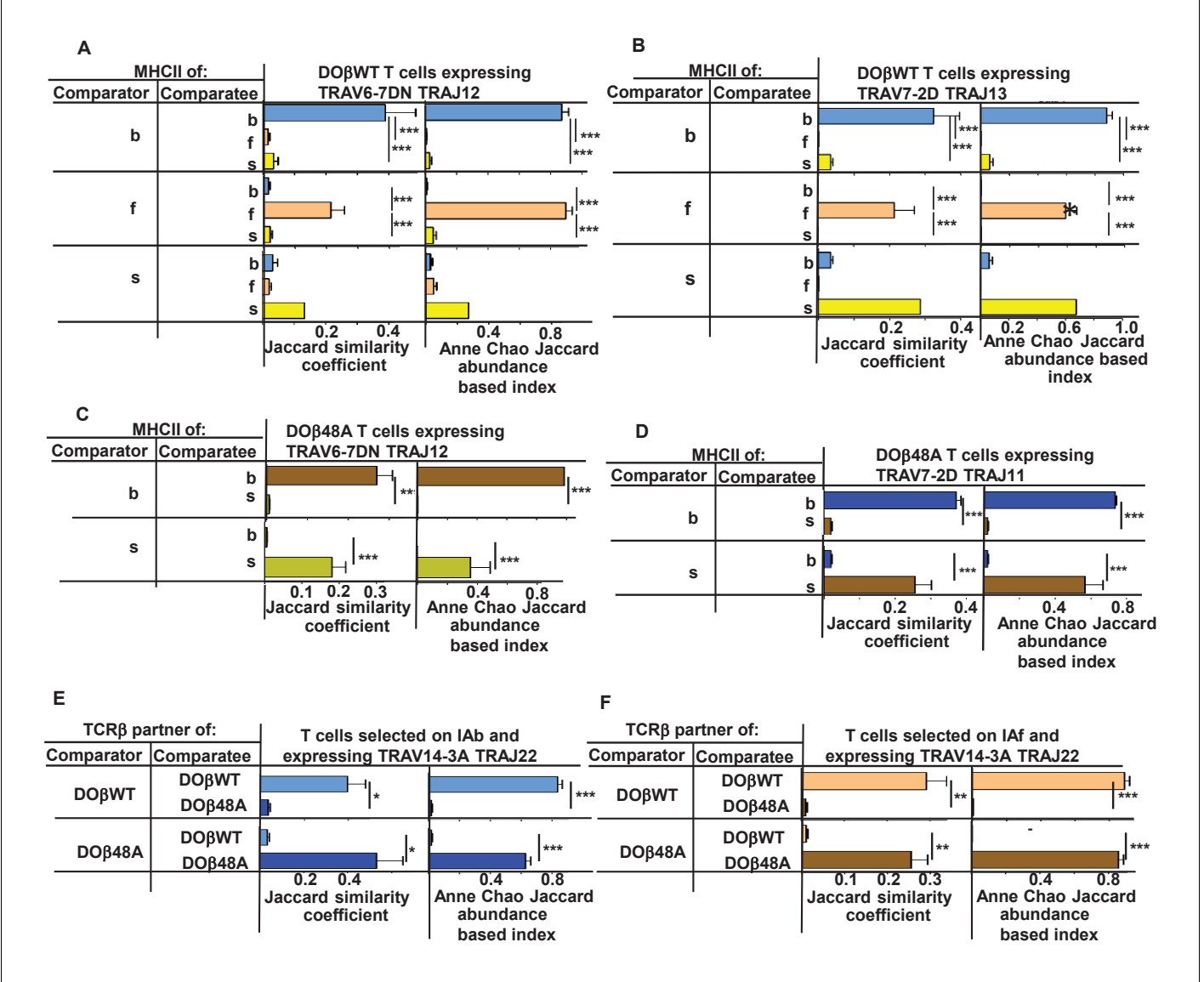

**Figure 8.** N region amino acids in CDR3α of naïve CD4 T cells are determined by the selecting MHCII allele and the co-selected TCRβ. TCRαs on naïve CD4 T cells of mice expressing a single TCRβ and various MHCII alleles were sequenced as described in *Figures 3* and *4*. Mice were as listed in those Figures. Comparisons were made of N regions derived from the same TRAV TRAJ pair providing that all mice in the comparisons expressed at least five different sequences involving the chosen TRAV TRAJ pair. Results shown are the means ± SEMs of the data from identical mice. Statistical analyses involved one way ANOVA tests with Newman-Keuls post test analyses (**A, B**) or Student t tests (**C–F**). *p<0.05, **p<0.01, ***p<0.001.
DOI: https://doi.org/10.7554/eLife.30918.019

The following figure supplements are available for figure 8:

**Figure supplement 1.** N region amino acids in CDR3α of naïve CD4 T cells are determined by the selecting MHCII allele and the co-selected TCRβ.
DOI: https://doi.org/10.7554/eLife.30918.020

**Figure supplement 2.** The frequency of amino acid use in CDR3α use on naïve CD4 T cells depends on the selecting MHCII allele, the coselected TCRβ and the length of the CDR3α.
DOI: https://doi.org/10.7554/eLife.30918.021

between the developing thymocyte and MHC proteins to which the cell is exposed in the thymus cortex (*Sprent et al., 1988*). However, the peptides presented to immature T cells in the thymus are also controlled by the allele of MHC involved. Thus, the MHC allele specificity of positive selection might be dictated by TCR contact with the MHC-engaged peptides rather than the MHC protein itself. If this were the case, positive selection might be dominated by the portion of TCRs that most

consistently engages the peptides bound to MHC, CDR3 sequences of TCRs, rather than the germ line encoded TRAVs and TRBVs. This idea is supported by the fact that, in some cases, peptides related to the activating antigen can stimulate positive selection of thymocytes bearing particular TCRs (*Ashton-Rickardt et al., 1994*; *Sebzda et al., 1994*; *Hogquist et al., 1994*; *Kraj et al., 2001*; *Smyth et al., 1998*). Moreover MHC proteins that were supposed to differ only in amino acids that bind peptide and that don't contact TCRs nevertheless were found to differ in their ability to select thymocytes bearing certain TCRs (*Nikolić-Zugić and Bevan, 1990*).

During positive selection could TCRs detect allelic differences between MHCs directly? Although that MHC amino acids that contact TCRs are quite well conserved (*Bjorkman et al., 1987*) they do vary somewhat between alleles (*Reche and Reinherz, 2003*). For the MHCII alleles studied here, the amino acids pointing towards the TCR are, at most positions, uniform, but $IA^b$ differs from $IA^f$ and $IA^s$ with a two amino acid insertion on the surface of its beta chain alpha helix, an insertion that causes an allele specific bulge in this helix. $IA^s$ also differs from the other MHCII alleles we studied with alpha chain H68Y and beta chain R70Q changes and position 72 of the MHCIIα alpha helix is a V in $IA^b$, an I in $IA^f$ and $IA^s$. Therefore the solvent exposed residues on the alpha helices of the MHCII proteins themselves could contribute to allele specific positive selection. These amino acids are contact points, not only for the CDR1 and CDR2 loops of TCRs, but also, sometimes for the TCR CDR3 regions. For example, in the structure on a TCR bound to the complex of $IA^u$ bound to a myelin basic protein, CDR3α engages polymorphic amino acids at positions 55 and 81 of the $IA^u$ alpha chain and CDR3β engages polymorphic amino acids at positions 65 of the $IA^u$ alpha chain and positions 67 and 70 of the $IA^u$ beta chain (*Maynard et al., 2005*).

The findings presented previously (*Merkenschlager et al., 1994*) and here, demonstrate that the allele of MHCII involved in positive selection affects the frequencies with which TRAV and TRAJ elements are selected and, most dramatically, the CDR3α sequences that appear on mature T cells, as previously indicated for CD8 T cells (*Ferreira et al., 2006*). The N terminal portion of CDR3α is provided by the TRAV, so the effects of MHCII on TRAV choice by a particular TCRα could actually be due to demands placed on the CDR3α rather than on the TRAV itself. On the other hand, since the CDR3α sequence is affected in part by its co-expressed TRAV, the demands of positive selection could be entirely on the TRAV, not the CDR3α. We think it likely that all three of the TCRα CDRs can play a role in positive selection. However, clearly CDR3α is involved since the sequences in the center of this element vary depend on the selecting MHCII allele, even if the accompanying TRAV and TRAJ are the same (*Figure 8*).

Overall, the results strongly suggest that positive selection allele specificity involves recognition of both MHC and peptide (reviewed in (*Klein et al., 2014*; *Vrisekoop et al., 2014*). In fact, given the geometry with which TCRs engage their MHC/peptide ligands, it is difficult to imagine that this would not be the case.

The data here also show that the sequence of the TCRβ chain affects the TCRα and CDR3α that can participate in positive selection almost as much as the selecting MHC allele does. Not only does the TCRβ affect which TCRαs and which CDR3αs will be successful, it also determines how many different TCRαs can do the job since, regardless of the MHCII allele involved, fewer TCRαs can be selected with DOβ48A than with DOβWT. These results are similar to those observed earlier that showed that fewer CD4 T cells are selected in DOβ48A-expressing versus DOβWT-expressing mice. Together these suggest that DOβ lacking an important MHC contact amino acid, the 'Y' at position 48, places more stringent requirements on TCRα for successful thymus selection (*Scott-Browne et al., 2009*).

Overall, the results show that the entire TCR sequence plays a role in positive selection. How can this be, given that selection is thought to occur during low affinity reactions? Naively one might have predicted that relatively few TCR-to-MHC/peptide interactions would be needed to reach the needed energy of interaction and these could be provided by just a portion of the TCR, not the entire molecule as suggested here. Some TCR configurations may interfere with contact with MHC/peptide or prevent the proper engagement of CD4 or CD8. Other TCR configurations may react too strongly with their ligand, leading to negative selection. This idea may apply to up to 70% of all TCRs (*Ignatowicz et al., 1996*; *Stritesky et al., 2013*). Competition for selecting ligands may also play a role. Also to be bourne in mind is the fact that here we are observing the consequences of many TCR selection events, some TCRs may be selected based on their TRAVs, others via their

TCRβs, with the observed results showing biases by both of these. Nevertheless the detrimental effects of an inappropriate CDR3α cannot be overcome by other elements of TCRα.

There are problems with the notion that the bound peptide is a determinant of MHC allele specific positive selection. Most notably, the fact that mature T cells, after selection on a single MHCII allele bound to a single peptide can respond to peptides that are unrelated in sequence to the selecting peptide (*Pawlowski et al., 1996*; *Ignatowicz et al., 1997*; *Nakano et al., 1997*; *Ebert et al., 2009*; *Lo et al., 2009*). Moreover, teleologically, the idea that the selecting peptides in the thymus are the only feature that governs T cell specificity doesn't seem evolutionarily favorable. Such might limit the ability of T cells to respond to foreign peptides that are unrelated to those in the thymus. Nevertheless, self peptides might provide an advantage anyway, by supporting the survival of mature T cells and also, perhaps, T cell responses to unrelated peptides when the self and foreign peptides are presented on the same cells (*Kirberg et al., 1997*; *Wülfing et al., 2002*). However, MHC-bound peptides on thymus cortical epithelial cells are not necessarily the same as those on peripheral cells (*Honey et al., 2002*; *Murata et al., 2007*) so this advantage may not be available for all T cells. It has also been suggested that positive selection on self peptides, by selecting for TCRs that react with peptides that are similar to self, might protect hosts from infection by organisms that have mutated such that their proteins resemble self. If negative selection were the only criterion for thymocyte maturation, invading organisms might be able to avoid recognition by T cells. Positive selection on self prevents such evasion (*Forsdyke, 2015*; *Vrisekoop et al., 2014*)

In the studies presented here, the total number of TCRαs that can be selected with a single TCRβ ranges between about 4600 and 30,000, depending on the selecting MHCII allele and partner TCRβ. Are these numbers surprisingly low or high? Based on the estimated numbers of TCRβs and TCRαs that appear in the periphery of an individual, it has previously been estimated that each TCRβ chain can be successfully selected with up to 25 different TCRαs (*Arstila et al., 1999*; *Casrouge et al., 2000*). Yet here, and in previous studies, it appears that, for a single TCRβ, the number of possible TCRα partners is at least 3 orders of magnitude larger. How to account for this large disparity? Probably it is caused by competition for selection in the thymus, a phenomenon that has been previously demonstrated (*Martins et al., 2014*; *Visan et al., 2006*). In a wild type thymus, each of the immature thymocytes is competing with a huge number of others bearing disparate TCRβ and TCRα sequences. In the thymus of a mouse expressing a single TCRβ, the immature thymocytes bear the same large number of TCRαs, and now all those expressing an even approximately suitable TCRα have the opportunity to be positively selected. This idea may be related to the profound bias towards distal TRAJs reported here, and therefore a predicted increased time available for rearrangements for the thymocytes in single TCRβ transgenic animals.

## Materials and methods

### Key resources table

| Reagent type (species) or resource | Designation | Source or reference | Identifiers | Additional information |
|---|---|---|---|---|
| T cell receptor beta chain from the DO11.10 hybridoma (mus musculus musculus) | DObWT | Haskins, K., Kubo, R., White, J., Pigeon, M., Kappler, J. and Marrack, P. The major histocompatibility complex-restricted antigen receptor on T cells. I. Isolation with a monoclonal antibody. J. Exp. Med. 157:1149, 1983. | | |
| T cell receptor beta chain from the DO11.10 hybridoma (mus musculus musculus) with the amino acid at position 48 of the chain mutated from a tyrosine to an alanine | DOb48A | 375. Scott-Browne, J.P., White, J., Kappler, J.W., Gapin, L. and Marrack, P. Germline-encoded amino acids in the abT cell receptor control thymic selection. Nature 458:1043–1046, 2009. PMC2679808 | | |
| C57BL/6J (mouse) | B6 | The Jackson Laboratory | | |
| Mice congenic with B6 but expressing H2f | | This publication | | |
| Mice congenic with B6 but expressing H2s | | This publication | | |

*Continued on next page*

*Continued*

| Reagent type (species) or resource | Designation | Source or reference | Identifiers | Additional information |
|---|---|---|---|---|
| Mice with one TCRa gene inactivated | a+/- | The Jackson Laboratory; Mombaerts et al. Nature 360:225–231,1992 | | |
| B6 mice with the TCRb genes inactivated | b-/- | The Jackson Laboratory; Creation of a large genomic deletion at the T-cell antigen receptor beta-subunit locus in mouse embryonic stem cells by gene targeting. Mombaerts et al. PNAS 88 3084–7, 1991 | | |
| Software for analysis and correction of TCRalpha sequences | | https://www.nationaljewish.org/research-science/programs-depts/biomedical-research/labs/kappler-marrack-research-lab/protocols | | |
| B6 TRAV and TRAJ sequences | MOVA-B6.VDB MOJA.JDB | https://www.nationaljewish.org/research-science/programs-depts/biomedical-research/labs/kappler-marrack-research-lab/protocols | | |
| TCRa sequences used in analyses | | | GEO accession GSE105129 | |

## Mice

Mice were purchased from the Jackson Laboratory, Bar Harbor ME and subsequently interbred in the Biological Research Center at National Jewish. Plasmids coding for the DO11.10 TCRβ chain (DOβWT) or its mutant, in which the tyrosine at position 48 was replaced by an alanine (DOβ48A) were created, with the human CD2 promoter to drive expression of the genes (*White et al., 1983*; *Greaves et al., 1989*). DNAs coding for the promoters and genes were injected into fertilized C57BL/6J (B6) eggs at the Mouse Genetic Core Facility at National Jewish Health. Mice produced from these eggs were crossed with animals lacking functional TCRβ genes (*Mombaerts et al., 1991*) and with B10.M (H2f) or B10.S (H2s) animals to create animals expressing the transgenic TCRβ genes, no other TCRβgenes and H2b, f or s. By similar intercrosses animals were produced that expressed no functional TCRα or TCRβ genes and H2b, f or s. These animals were intercrossed to give rise to animals expressing either DOβWT or DOβ48A, no other TCRβ genes, TCRα+/- and H2 b, f or s. Animals were subsequently used for analysis if they expressed the TCRα locus derived from B6 rather than B10 animals.

Animals were handled in strict accordance with good animal practice as defined by the relevant national and/or local animal welfare bodies, and all animal work was approved by the National Jewish Health Animal Care and Use Committee (IACUC). The protocol was approved by National Jewish IACUC (protocol number AS2517).

## T cell isolation

Cells were isolated from the thymuses, spleens (B6 analyses) or peripheral lymph nodes (DOβWT or DOβ48A analyses) of 6–14 week old mice. CD4 T cells were isolated by negative selection on MACS columns (Milltenyi Biotech). The cells were stained with antibodies conjugated to a fluorochrome and specific for: Pacific Blue-CD4 (RM4-5, BioLegend, 100531), Alexa488-TCRβ(Ham-597, made in house), PE-B220 (RA3-6B2, BD Pharmingen, 553090), PE-TCRδ(GL3, BD Pharmingen 553178), PE-CD8α (53–6.7, BD Pharmingen, 553033), PE-Cy5-CD25 (PC61.5, eBioscience, 15-n251-82), Alexa647-CD44 (made in house), PE-Cy7-CD62L (MEL-14, eBioScience, 25-0621-82). The cells were sorted based on their expression of CD4, TCRβ, low levels of CD44 and high levels of CD62L and absence of staining with PE and PE-Cy5. Cells were sorted into staining buffer (BSS, 2% fetal bovine serum, plus sodium azide) by a MoFloXDP (Beckman Coulter Life Sciences or Synergy SY3200 (iCyt) instruments at the National Jewish Health Flow Cytometry Core Facility.

## Retroviral infection of T cells

Retroviruses expressing DOβWT or DOβ48A and green fluorescent protein were produced as described in *Scott-Browne et al. (2009)*. CD4 T cells were purified, by negative selection on Auto-max columns, from the spleens and lymph nodes of DOβ48A transgenic, TCRβ-/- mice expressing

various MHC alleles. The cells were activated by 24 hr culture on plates pre-coated with anti-TCRβ (Ham597) and anti-CD28 (37.51) The supernatants were then removed from the plates and replaced with supernatants containing the DOβWT or DOβ48A retroviruses and 8 ug/ml polybrene in culture medium. The cells were spun at 2000G in bags containing $10\%CO_2/90\%air$ at 37.C for 2 hr. At this point the medium was replaced with complete culture medium containing 10% fetal bovine serum and cultured for 1d followed by addition of IL-2. Three days later the cells were harvested and challenged as described below.

## Assessment of MHC reactivity of transduced T cells

Red blood cell depleted spleen cells from mice expressing various MHC alleles were cultured overnight with IL-4 plus GM-CSF. The cells were then thoroughly washed and used, at a dose of $10^6$ cells/well, to stimulate $10^6$/well TCRβ transduced CD4+ T cells, prepared as described above. These wells were cultured for 51/2 hr in a final total volume/well of 200 ul of CTM. The cells were then fixed (Permafix), stained and analyzed for expression of CD4 (PerCP anti-CD4), GFP and CD69 (PE anti-CD69).

## TCRα sequencing and analysis

RNA was isolated from purified naïve CD4 T cells, PCR'd to expand *Tcra* sequences and sequenced as described in *Silberman et al. (2016)*. Post-sequencing analysis was performed to identify the *Trav* and *Traj* genes for each sequence along with its corresponding CDR3. *Trav* family and subfamily members were assigned based on the IMGT designations with modifications based on our own analysis of expressed TRAV sequences in B6 mice. IMGT has identified two gene duplication events in the B6 *Trav* locus, the 'original' genes, most of which are closest to the TRAJ locus are designated by their family number and a number indicating their subfamily membership. Here, for ease of analysis, we have added the letter 'A' to their designation, eg TRAV01-1A. TRAV subfamily members in the IMGT designated duplicated 'D' and new 'N' genes we add the letters 'D' or 'N', eg TRAV07-6D or TRAV07-6N. In some cases the entire nucleotide sequences of subfamily members are identical and, therefore, indistinguishable by our analyses. In these cases the subfamily members are designated to include all possible source genes, eg TRAV06-3ADN or TRAV06-6AD.

Errors occur during sequencing reactions and accumulate as the numbers of sequences acquired increase (*Bolotin et al., 2012*; *Liu et al., 2014*). The sequences were all corrected for errors in the *Trav* and *Traj* elements, which do not somatically mutate. However, because the amino acids in and flanking the non germ line encoded portions of CDR3 regions could not be corrected, sequences with errors in these elements are bound to appear at some low frequency and cause a gradual rise in the species accumulation curves. To eliminate these misreads we decided to include in our analyses only those TCRαsequences that occurred more than once in each sample. To correct for sequencing errors within the CDR3, the sequences were modified by replacing erroneous nucleotides with the appropriate germline-encoded nucleotides whenever a discrepancy was observed. Such correction was possible only when a nucleotide difference could be resolved by aligning to the germline *Trav* and/or *Traj* genes. To avoid making inappropriate changes to the potentially non germline encoded portions of CDR3α, such corrections were applied only if the change from the germline sequence occurred more than three nucleotides before the predicted end of the *Trav* genes or more than three nucleotides after the predicted end of the *Traj* gene. Finally, the amino acid usage within the CDR3α was determined for each sequence to identify any patterns in the CDR3 regions in sequences belonging to T cells from one MHC haplotype versus another. All of the analysis was performed using in-house programs developed in Python 2.7. Software and sequences used to analyze and correct TCR alpha sequences are at the lab webpage https://www.nationaljewish.org/research-science/programs-depts/biomedical-research/labs/kappler-marrack-research-lab/protocols or available on request to PM, SHK or JWK. The raw and analyzed sequences used in this paper are at GEO accession GSE105129.

In order to represent the differential *Trav* and *Traj* gene usage in TCRs sequenced from different mouse samples, we used edgeR from the R/Bioconductor package. A threshold of $p < 0.05$ was used to identify genes that were most significantly differentially expressed between samples.

Euclidean distances for TRAVs and TRAJs were calculated as $\log_2$ transformed counts per $10^4$ sequences.

## Acknowledgements

The authors thank Dean Becker for production of the transgenic mice, Desiree Garcia and Tabitha Turco for help in mouse breeding, Shirley Sobus and Josh Loomis of the National Jewish Health Cytometry Core and Randi Anselment for the TCRα sequencing. This work was supported in part by NIH grants AI-18785 (PM) AI-092108 (LG), AI-103736 (LG), DK-099317 (MN) and T32 AI-006405.

## Additional information

### Funding

| Funder | Grant reference number | Author |
|---|---|---|
| National Institutes of Health | AI 18785 | Philippa Marrack |
| National Institutes of Health | AI 092108 | Laurent Gapin |
| National Institutes of Health | AI 103736 | Laurent Gapin |
| Howard Hughes Medical Institute | | John Kappler |
| National Institutes of Health | DK099317 | Maki Nakayama |

The funders had no role in study design, data collection and interpretation, or the decision to submit the work for publication.

### Author contributions

Philippa Marrack, Conceptualization, Data curation, Formal analysis, Funding acquisition, Investigation, Writing—original draft, Project administration, Writing—review and editing; Sai Harsha Krovi, Software, Formal analysis, Investigation, Methodology, Writing—review and editing; Daniel Silberman, Formal analysis, Investigation, Methodology; Janice White, Maki Nakayama, Data curation, Investigation, Methodology; Eleanor Kushnir, Randy Anselment, Investigation, Methodology; James Crooks, Sonia Leach, Data curation, Formal analysis; Thomas Danhorn, Data curation, Software, Formal analysis; James Scott-Browne, Conceptualization, Data curation, Software, Formal analysis, Writing—review and editing; Laurent Gapin, Conceptualization, Data curation, Formal analysis, Supervision, Investigation, Methodology, Writing—review and editing; John Kappler, Conceptualization, Data curation, Software, Formal analysis, Funding acquisition, Methodology, Writing—review and editing

### Author ORCIDs

Philippa Marrack http://orcid.org/0000-0003-1883-3687
Thomas Danhorn http://orcid.org/0000-0002-3861-8602

### Ethics

Animal experimentation: This study was performed in strict accordance with the recommendations in the Guide for the Care and Use of Laboratory Animals of the National Institutes of Health. All of the animals were handled according to approved institutional animal care and use committee (IACUC) protocols (AC-2517) of National Jewish Health. The protocol was approved by the Institutional Animal Care and Use Committee of National Jewish Health. Every effort was made to minimize suffering.

### Decision letter and Author response

Decision letter https://doi.org/10.7554/eLife.30918.027
Author response https://doi.org/10.7554/eLife.30918.028

# Additional files

## Supplementary files

• Supplementary file 1. In normal mice, a significant number of TCRα sequences appear on naïve CD4 T cells regardless of the selecting MHCII allele. Naïve CD4 T cells were isolated from the lymph nodes of normal mice of the indicated strains and their TCRα sequences identified as described in the Materials and methods section. Shown are the %s of unique sequences and the %s of total sequences that were shared between pairs of mice of the indicated strains. Data were obtained from three independently sequenced B6 mice and one each B6.AKR and B6.NOD animals and are the means and standard errors of the means of the comparisons.
DOI: https://doi.org/10.7554/eLife.30918.022

• Supplementary file 2. Sequences of TCRβ transgenes
DOI: https://doi.org/10.7554/eLife.30918.023

• Transparent reporting form
DOI: https://doi.org/10.7554/eLife.30918.024

## Major datasets

The following dataset was generated:

| Author(s) | Year | Dataset title | Dataset URL | Database, license, and accessibility information |
|---|---|---|---|---|
| Marrack P, Krovi SH, Silberman D, White J, Kushnir E, Nakayama M, Crook J, Danhorn T, Leach SM, Anselment R, Scott-Browne J, Gapin L, Kappler JW | 2017 | The somatically generated T cell receptor CDR3a contributes to the MHC allele specificity of the T cell receptor | https://www.ncbi.nlm.nih.gov/geo/query/acc.cgi?acc=GSE105129 | Publicly available at the NCBI Gene Expression Omnibus (accession no: GSE105129) |

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
