## [Decision Letter]

Thank you for submitting your article "The somatically generated T cell receptor CDR3a contributes to the MHC allele specificity of the T cell receptor" for consideration by *eLife*. Your article has been favorably evaluated by Arup Chakraborty (Senior Editor) and three reviewers, one of whom is a member of our Board of Reviewing Editors.. The following individual involved in review of your submission has agreed to reveal their identity: Nilabh Shastri (Reviewer #2).

The reviewers have discussed the reviews with one another and the Reviewing Editor has drafted this decision to help you prepare a revised submission.

You reveal new insights into the mechanisms regulating positive selection of T-cell receptors (TCRs) by MHC class II molecules. You show that the bias in the specificity of TCRs for self-MHC expressed in the thymus results not only from germ-line encoded elements of the TCR α chain, but also from its somatically generated CDR3 region. The findings help explain the structural basis for positive selection of T cell receptors in the thymus that is known to be skewed by thousands of polymorphic MHC alleles.

A clever approach is used to analyze the structural features of the TCR compatible with positive selection, transgenic mice that express the β chain (DOβWT) of the TCR isolated from a T cell hybridoma specific for either the Ad or Ab MHC class II molecules presenting the same OVA-derived peptide. In these mice, due to allelic exclusion, all T cells express the same β TCR transgene, but the T cells are free to pair with any α TCR chain. They compare TCRs selected in these transgenic mice with another transgenic mouse produced using the same β TCR (DOβ48A) with a mutation (Y48A) known to reduce its ability to interact with MHC II molecules and thus serve as a control for MHC specific factors. These two transgenes were then bred onto mice expressing either Ab, Af or As MHC class II molecules. As expected, the TCRs of CD4 T cells selected by each of these MHC II alleles were distinct in their bias for the self-MHC molecule.

Large-scale careful and statistically-sound analysis of the TCR α chains was then carried out among the CD4 T cells in the various mice. To ensure that only a single α chain was expressed by the T cells mice were bred onto the TCR β-/- homozygous and TCR α +/- heterozygous backgrounds. Extensive analysis of TCRs expressed by pre-selected versus naïve CD4 T cells showed that the selection of TCR α chains was clearly dependent upon the MHC II allele expressed in the thymus. Furthermore, the selected α chains showed clear impact of both the germline (CDR1 and CDR2) as well as somatically generated CDR3 containing the V-J junctions.

This is a nicely written paper. You are commended for presenting a thoughtful commentary of your well-controlled and carefully executed experiments and an excellent discussion of relevant issues of the role of MHC versus peptide residues in the positive selection interactions with features of the TCR. Although this study, like others, cannot distinguish whether the observed changes are caused by unique features of the MHC or the peptide, it advances the field by providing a new perspective. The study's novelty and importance lie in the characterization of the impact of MHC alleles on thymic selection of PAIRS of TCRαβ chains, not just TCRα or TCRβ-so they are really characterizing the impact of MHC alleles on the antigen-binding TCR repertoire. The results are clearly presented (which is truly challenging for this dataset) and persuasively discussed. Overall, the results strongly support the notion that the presented self peptides govern the allele bias of positive selection.

One could ask what would be observed in F1 mice, but this is likely beyond what is feasible within the scope of this study. The manuscript as is presents an appropriate amount of work, and is very definitive. Appropriate quantitative tools are employed to characterize the repertoire. Controls are employed at multiple steps that ensure the quality of the sequencing data. Overall, we found very little lacking in this report and have only minor suggestions to improve the manuscript.

Please follow the following recommendations for revision:

1) Paragraph 1 should include citations to relevant literature.

2) Relevant to point 1 above, it would be informative to include a discussion of models for TCR recognition of MHC-peptide complexes. For example, when the first crystal structures of TCR-pMHC complexes were solved (note that Garboczi et al., 1996 should be cited along with Garcia et al., 1996), they revealed TCRs that were aligned diagonally across the pMHC such that the prevailing model at the time (CDRs 1 and 2 recognize the MHC helices and the more diverse CDR3s would recognize the bound peptide) had to be rethought. Subsequent structures showed variability in TCR orientation on MHC/peptide complexes, ranging from close to the perpendicular orientation predicted by the above-described model (predicting that CDR3alpha would interact only with the bound peptide) to the diagonal orientations in which CDR3alpha could interact with the MHC helices as well as the bound peptide (nicely reviewed in Hennecke and Wiley, 2001, Cell 104:1-4). Thus for a TCR that orients diagonally across a pMHC, it would not be surprising that changes in CDR3alpha could affect positive selection of TCRs.

3) Figure 5—figure supplement 2 is difficult to understand. Perhaps the legend could explain more fully what exactly is plotted, and what results are meaningful (when points are skewed off the line).

4) We agree with the authors in their final point of the Discussion – that the strong bias for distal Js and the large number of paired V's does seem to suggest that the TCRβ is "desperate" in TCRβ transgenic mice. But distal Js are not any more likely to generate selectable CDR3s than proximal, I would imagine. So I wonder if the overall threshold of selection ends up being changed in TCRβ transgenic mice. Have the authors examined the level of Nur77GFP in CD4 T cells in mice on a non-Tg versus TCRβ transgenic background?

5) The TCR sequences should be deposited in appropriate databases.

6) Nomenclature: is it misleading to refer to the CDR3 loops as "somatically generated" given that at least some of the residues are germline encoded within the J (for Vα) and D plus J (for Vβ) gene segments?

7) The rigor and extent of statistical information is appropriate, although it could be argued that the data are better visualized using error bars that indicate standard deviation, rather than standard error of the mean. P values are derived with appropriate statistical tools.

---

## [Author Response]

1) Paragraph 1 should include citations to relevant literature.

We have added appropriate references to the statements made in paragraph 1 of the manuscript.

2) Relevant to point 1 above, it would be informative to include a discussion of models for TCR recognition of MHC-peptide complexes. For example, when the first crystal structures of TCR-pMHC complexes were solved (note that Garboczi et al., 1996 should be cited along with Garcia et al., 1996), they revealed TCRs that were aligned diagonally across the pMHC such that the prevailing model at the time (CDRs 1 and 2 recognize the MHC helices and the more diverse CDR3s would recognize the bound peptide) had to be rethought. Subsequent structures showed variability in TCR orientation on MHC/peptide complexes, ranging from close to the perpendicular orientation predicted by the above-described model (predicting that CDR3alpha would interact only with the bound peptide) to the diagonal orientations in which CDR3alpha could interact with the MHC helices as well as the bound peptide (nicely reviewed in Hennecke and Wiley, 2001, Cell 104:1-4). Thus for a TCR that orients diagonally across a pMHC, it would not be surprising that changes in CDR3alpha could affect positive selection of TCRs.

In the Introduction to the manuscript we have changed our description of the controversies over the orientation of TCRs on MHC, controversies that were eventually resolved by Xray crystallographic solution of the structures of TCR/MHC complexes. We have added the Garboczi et al. and Hennecke and Wiley references, that were suggested by the reviewers, as well as several other appropriate references. We agree that the structural studies suggest that the CDRs loops might affect positive selection, by recognizing either MHC and/or peptide and have tried to make that point more clearly.

3) Figure 5—figure supplement 2 is difficult to understand. Perhaps the legend could explain more fully what exactly is plotted, and what results are meaningful (when points are skewed off the line).

We have revised the legend to Figure 5—figure supplement 2 (and Figure 6—figure supplement 1) to make the figures easier to understand. We hope we have succeeded.

4) We agree with the authors in their final point of the Discussion – that the strong bias for distal Js and the large number of paired V's does seem to suggest that the TCRβ is "desperate" in TCRβ transgenic mice. But distal Js are not any more likely to generate selectable CDR3s than proximal, I would imagine. So I wonder if the overall threshold of selection ends up being changed in TCRβ transgenic mice. Have the authors examined the level of Nur77GFP in CD4 T cells in mice on a non-Tg versus TCRβ transgenic background?

We thank the reviewers for the suggestion that we take a look at Nur77 expression in selecting thymocytes. Therefore we used intracellular staining to measure the levels of Nur77 and Egr2 in CD69+ TCR+ thymocytes undergoing selection in normal C57BL/6 mice and the DOβWT and DOβ48A mice (3 of each) that were the subjects of the studies in our manuscript. We found that Nur77 was at slightly (statistically significantly) lower levels in thymocytes undergoing selection in the DOβWT and DOβ48A mice by comparison with the same kinds of cells in B6 mice. Levels of Egr2 were also lower in the TCRβ transgenic mice, but this was only statistically significant in the DOβ48A animals. (see Table below) This is an interesting but subtle result that will require extensive experiments to follow up, so we haven’t included the result in the current manuscript, simply mentioned the idea as a possibility.

Nur77 levels are lower in positively selected thymocytes from TCRb transgenic mice than in the equivalent thymocytes in B6 animals.Average +/- SE Geometric mean levels in CD69high TCRhigh thymocytesB6DOβWTDOβ48ANur77823.0 +/- 34.1675.0 +/- 28.9*600.3 +/- 39.6*Egr21055.0 +/- 20.9951.3 +/- 122.8777.0 +/- 52.4*** p<.05 by comparison with B6** p<.01 by comparison with B6

5) The TCR sequences should be deposited in appropriate databases.

The sequences are now deposited in GEO accession number GSE105129. The token for reviewer access to this site is gjwvoyigzpwxnof. The programs we used to analyze the results are at our webpage, (https://www.nationaljewish.org/research-science/programs-depts/biomedical-research/labs/kappler-marrack-research-lab/overview) under the Protocols tab and are also available by contacting JWK or PM by email. These sites are mentioned in the Materials and methods section of the current manuscript.

6) Nomenclature: is it misleading to refer to the CDR3 loops as "somatically generated" given that at least some of the residues are germline encoded within the J (for Vα) and D plus J (for Vβ) gene segments?

Thank you for suggesting this correction. We agree and have changed the title and text of the manuscript such that the reference to CDR3 loops now reads as “somatically generated portion of CDR3α”.

7) The rigor and extent of statistical information is appropriate, although it could be argued that the data are better visualized using error bars that indicate standard deviation, rather than standard error of the mean. P values are derived with appropriate statistical tools.

We would prefer to leave our figures showing standard errors because such represents the likelihood that any given value we are showing is correct. The magnitude of the standard deviations is instead represented by the p values, since these are based on the spread of values.